# Humoral Response Kinetics and Cross-Immunity in Hospitalized Patients with SARS-CoV-2 WT, Delta, or Omicron Infections: A Comparison between Vaccinated and Unvaccinated Cohorts

**DOI:** 10.3390/vaccines11121803

**Published:** 2023-12-01

**Authors:** Hyunhye Kang, Jihyun Lee, Jin Jung, Eun-Jee Oh

**Affiliations:** 1Department of Laboratory Medicine, Seoul St. Mary’s Hospital, College of Medicine, The Catholic University of Korea, Seoul 06591, Republic of Korea; hyunhye@cmcnu.or.kr (H.K.); jiinj@catholic.ac.kr (J.J.); 2Research and Development Institute for In Vitro Diagnostic Medical Devices, College of Medicine, The Catholic University of Korea, Seoul 06591, Republic of Korea; 3Department of Biomedicine & Health Sciences, Graduate School, The Catholic University of Korea, Seoul 06591, Republic of Korea; onion1002@catholic.ac.kr

**Keywords:** SARS-CoV-2, vaccine, humoral immunity, neutralization, variant

## Abstract

With the ongoing evolution of severe acute respiratory virus-2 (SARS-CoV-2), the number of confirmed COVID-19 cases continues to rise. This study aims to investigate the impact of vaccination status, SARS-CoV-2 variants, and disease severity on the humoral immune response, including cross-neutralizing activity, in hospitalized COVID-19 patients. This retrospective cohort study involved 122 symptomatic COVID-19 patients hospitalized in a single center. Patients were categorized based on the causative specific SARS-CoV-2 variants (33 wild-type (WT), 54 Delta and 35 Omicron) and their vaccination history. Sequential samples were collected to assess binding antibody responses (anti-S/RBD and anti-N) and surrogate virus neutralization tests (sVNTs) against WT, Omicron BA.1, and BA.4/5. The vaccinated breakthrough infection group (V) exhibited higher levels of anti-S/RBD compared to the variant-matched unvaccinated groups (UVs). The Delta infection resulted in a more rapid production of anti-S/RBD levels compared to infections with WT or Omicron variants. Unvaccinated severe WT or Delta infections had higher anti-S/RBD levels compared to mild cases, but this was not the case with Omicron infection. In vaccinated patients, there was no difference in antibody levels between mild and severe infections. Both Delta (V) and Omicron (V) groups showed strong cross-neutralizing activity against WT and Omicron (BA.1 and BA.4/5), ranging from 79.3% to 97.0%. WT (UV) and Delta (UV) infections had reduced neutralizing activity against BA.1 (0.8% to 12.0%) and BA.4/5 (32.8% to 41.0%). Interestingly, patients who received vaccines based on the ancestral spike exhibited positive neutralizing activity against BA.4/5, even though none of the study participants had been exposed to BA.4/5 and it is antigenically more advanced. Our findings suggest that a previous vaccination enhanced the humoral immune response and broadened cross-neutralizing activity to SARS-CoV-2 variants in hospitalized COVID-19 patients.

## 1. Introduction

The emergence of severe acute respiratory syndrome coronavirus 2 (SARS-CoV-2) has led to a global pandemic with a significant impact on public health. Vaccination has been a crucial strategy in combating the spread of the virus and reducing the severity of the coronavirus disease 2019 (COVID-19). It is now widely accepted that vaccination induces the production of binding antibodies against the SARS-CoV-2 spike protein, which neutralize the virus and protect against infection [1,2]. While vaccines have proven effective in preventing COVID-19 and reducing the severity of illness, the virus has shown the ability to evolve and give rise to variants of concern (VOC). These variants harbor mutations in the spike protein or other viral components, allowing them to evade the immune response or enhance their transmissibility [1,3,4].

The ongoing evolution of the virus and the emergence of new variants, along with vaccination and hybrid immunity [5,6,7,8], have led to the complex landscape of immunity surrounding COVID-19. Efforts have been made to investigate whether a particular SARS-CoV-2 variant plays a specific role in humoral immunity following COVID-19. Omicron, currently the most prevalent variant globally, demonstrated increased resistance to neutralization [9,10]. Previous studies have demonstrated increased neutralizing antibody titer in severe infections and breakthrough infections, compared to mild infections and primary infections, respectively [11,12,13]. However, the impact and relationship between different strains of SARS-CoV-2 variants, disease severity, and prior vaccination in eliciting humoral immunity, particularly cross-neutralizing activity, remains uncertain. Most importantly, it is crucial to ascertain whether vaccines targeting the ancestral SARS-CoV-2, on which most available vaccine platforms are built, effectively induce cross-neutralizing capabilities and continue to provide protection against VOCs and newly emerging strains.

In this study, we conducted a comprehensive analysis to examine the effects of previous vaccination, the causative variant, and disease severity on the kinetics of humoral immune response in a specific subgroup of hospitalized COVID-19 patients. Additionally, we evaluated the ability to generate cross-neutralizing activity against both the original SARS-CoV-2 strain and the Omicron variant.

## 2. Materials and Methods

### 2.1. Patients and Samples

We collected a total of 369 consecutive serum samples from 122 patients who were hospitalized due to symptomatic COVID-19 at Seoul St. Mary’s Hospital from February 2020 to June 2022. SARS-CoV-2 infection was confirmed by reverse transcriptase polymerase chain reaction (RT-PCR) in all the enrolled patients [14,15]. We obtained each patient’s medical records, including information on medical conditions, vaccination status, and any previous history of SARS-CoV-2 infection from the hospital’s electronic medical records. To assess the kinetics of binding antibody levels, we categorized the collected samples into four groups based on the number of days elapsed since the onset of symptoms: <7 days, 7–14 days, 15–21 days, and >21 days. The severity of the disease was classified as either mild or severe, as described in a previous study [12]. Briefly, patients classified as mild include those who had various signs and symptoms of COVID-19 but did not have shortness of breath, dyspnea, or abnormal chest imaging. Patients were classified as having severe disease if they had an SpO_2_ < 94% on room air, an arterial partial pressure of oxygen to fraction of inspired oxygen (PaO_2_/FiO_2_) ratio < 300 mm Hg, a respiratory rate > 30 breaths/min, or lung infiltrates > 50%. Serum samples were collected during the hospitalization between 1 and 8 times. The median interval between symptom onset and sampling was 13 days (range; 1–54 days). All serum samples were stored at 4 °C for up to two weeks and divided into aliquots for evaluation. Serum aliquots were subsequently stored at −80 °C until the time of analysis. This study was approved by the Institutional Review Board (IRB) of Seoul St. Mary’s Hospital (XC20SIDI0069). The need for written informed consent was waived by the IRBs due to the retrospective nature of the study and the use of residual serum samples.

### 2.2. Determination of SARS-CoV-2 Variants

We classified enrolled patients based on the specific SARS-CoV-2 variant likely responsible for their infection. During the sample collection, South Korea experienced three distinct waves of COVID-19, each characterized by a predominant variant. The first wave occurred from 16 January 2020 to 24 July 2021, when the ancestral SARS-CoV-2 variant was predominated. This was followed by the Delta-dominant period from 25 July 2021 to 15 January 2022. The third wave spanned from 16 January 2022 to 23 July 2022, during which the Omicron variants, specifically BA.1 and sub-variant BA.2, circulated [16]. A previous study showed that each of these variants was almost exclusively predominant in South Korea during their respective time periods [3,17]. In this study, each predominant variant is hereafter referred to as wild type (WT), Delta, and Omicron, respectively.

### 2.3. Measurement of SARS-CoV-2 Anti-S/RBD and Anti-N Antibody Levels

We analyzed patient serum samples using the Elecsys Anti-SARS-CoV-2 assay (Roche Diagnostics, Basel, Switzerland), which detects binding antibodies to the receptor-binding domain (RBD) of the spike protein (anti-S/RBD) and antibodies to the viral nucleocapsid (anti-N), according to the manufacturer’s instructions. Binding antibody was measured as units per mL (U/mL), and the traceable units of binding antibody per mL (BAU/mL) were calculated using conversion factors according to the WHO international standard for anti-SARS-CoV-2 immunoglobulin. To determine positivity, we employed the manufacturer’s recommended cutoff values of 0.8 U/mL for anti-S/RBD and 1.0 U/mL for anti-N.

### 2.4. SARS-CoV-2 Surrogate Virus Neutralization Test 

The SARS-CoV-2 surrogate virus neutralization test (sVNT) (GenScript cPassTM, Piscataway, NJ, USA) was performed as previously described [18]. In brief, patient samples, along with positive and negative controls, were diluted at a 1:10 ratio with a sample dilution buffer. These dilutions were then mixed with a horseradish peroxidase conjugated recombinant SARS-CoV-2 RBD solution and incubated at 37 °C for 30 min. The mixtures were then incubated for 15 min at 37 °C in a capture plate precoated with human angiotensin-converting enzyme 2 (hACE2) protein. After a washing step, tetramethylbenzidine (TMB) solution was added and the plate was incubated in the dark at room temperature for 15 min. A stop solution was added to quench the reaction and the absorbance was immediately read at 450 nm on an enzyme-linked immunosorbent assay (ELISA) microplate reader. The cutoff of ≥30% inhibition was applied according to the manufacturer’s instructions. Additionally, for samples collected more than 15 days after the onset of symptoms, we also conducted additional neutralization analyses against the Omicron BA.1 and sub-variants BA.4 and BA.5 (BA.4/5) using the variant targeting sVNT assays (GenScript).

### 2.5. Statistical Analysis

Continuous data, expressed as median with interquartile range or 95% confidence intervals, were compared by the Mann–Whitney U test. When multiple samples from the same patient were included in the same group, the median value was used as a representative measure. Categorical data are presented as counts and percentages, and data were compared with Fisher’s exact test. Data analysis and visualization were performed using Prism version 9.4.1 for Windows (GraphPad, San Diego, CA, USA) or MedCalc statistical software version 20.114 (MedCalc Software Ltd., Ostend, Belgium) and a two-tailed *p* value < 0.05 was considered statistically significant.

## 3. Results

### 3.1. Patient Characteristics

The study cohort consisted of patients who were unvaccinated (*n* = 69) or had received COVID-19 mRNA or adenovirus vector vaccines prior to infection (*n* = 53). The enrolled patients were categorized into five groups based on their vaccination status and variant exposure as follows: unvaccinated patients with Wild-Type (WT) infection [WT (UV)] (*n* = 33), unvaccinated patients with Delta infection [Delta (UV)] (*n* = 27), patients who experienced Delta breakthrough infections after vaccination [Delta (V)] (*n* = 27), unvaccinated patients with Omicron infection [Omicron (UV)] (*n* = 9), and patients who had Omicron breakthrough infections after vaccination [Omicron (V)] (*n* = 26) (Figure 1). The median duration from the last vaccination to breakthrough infection is 79 days (range 2–173) for Delta (V) and 126 days (range 23–294) for Omicron (V). The characteristics of each patient group are summarized in Table 1. The overall median age of enrolled patients was 68 years, with 41.0% (50/122) being female. Patients with underlying medical risk factors comprised 72.7% (24/33) in the WT (UV) group, 88.9% (24/27) in the Delta (UV) group, 88.9% (24/27) in the Delta (V) group, 88.9% (8/9) in the Omicron (UV) group, and 96.2% (25/26) in the Omicron (V) group. The most common medical risk factor was hypertension (35.2%, 43/122), followed by diabetes mellitus (24.6%, 30/122). Among the 53 patients who had breakthrough infections after vaccination, 27 had received two doses of the vaccine and were infected with the delta variant. The remaining 26 patients, including 18 who had received two vaccine doses and the rest who had received three doses, were infected with the Omicron variant. A total of 81 out of the 122 patients (66.4%) experienced severe disease, with 14 patients in the WT infection group, 43 in the Delta infection group, and 24 in the Omicron infection group. The proportions of severe infection in the WT, Delta, and Omicron variants did not differ to a statistically significant degree. Throughout the hospitalization stay, mortality rates were observed as follows: 1 out of 33 for WT (UV), 7 out of 27 for Delta (UV), 7 out of 27 for Delta (V), 3 out of 9 for Omicron (UV), and 11 out of 26 for Omicron (V). The 14 Omicron-infected patients were reported to have succumbed to underlying diseases rather than to COVID-19 itself.

### 3.2. Kinetics of Anti-S/RBD and Anti-N Antibody Responses in Studied Groups

When we monitored anti-S/RBD antibody levels in hospitalized patients after infection, we found significantly higher levels in the groups with vaccinated breakthrough infections compared to the unvaccinated infection group (Figure 2A,B). When comparing the median antibody levels in samples collected more than 21 days after infection with the same variant, the vaccinated group exhibited significantly higher levels compared to the variant-matched unvaccinated groups [Delta (V) showed a 126.8-fold increase compared to Delta (UV), while Omicron (V) displayed a 67.4-fold increase compared to Omicron (UV), *p* = 0.0035 and 0.0045, respectively].

Next, we compared the levels of anti-S/RBD antibodies between different variant groups (WT, Delta, and Omicron), categorizing samples into four groups based on the time elapsed since the onset of symptoms: <7 days, 7–14 days, 15–21 days, and >21 days. The Delta variant showed significantly higher levels of anti-S/RBD antibodies in samples collected between day 7 and day 21 post-infection, compared to WT and Omicron infections in both unvaccinated and vaccinated groups.

Regarding the anti-N antibody levels, there were no significant differences observed between the unvaccinated infection group and the vaccinated breakthrough infection groups (Figure 2C,D). When comparing the anti-N antibody levels between variants within the subdivided four groups, patients infected with the Delta variant showed significantly higher levels in samples obtained on days 15 to 21 post-infection compared to the WT (UV) and Omicron (V) groups.

### 3.3. Anti-S/RBD and Anti-N Antibody Responses in Studied Groups Stratified Symptom Severity

We examined whether disease severity had an impact on humoral immunity by comparing antibody levels based on prior vaccination, variant exposure, and disease severity (Figure 3).

In the unvaccinated infection group, severe infections with the WT or Delta variants showed higher antibody levels against S/RBD antigens compared to mild infections (*p* < 0.05). However, there was no difference in S/RBD antibody levels between the severe and mild unvaccinated Omicron infection groups (*p* > 0.05). Regarding the anti-N antibody, the overall trend indicated that antibody titers were higher in the severe disease groups, but only the Delta infection group showed a statistically significant difference (*p* < 0.05).

In the vaccinated infection group, there was no difference in antibody levels against S/RBD or N antigen between mild and severe infections in both the Delta and Omicron infection groups.

When comparing fatal cases with non-fatal individuals, the former exhibited higher anti-S/RBD levels (*p* = 0.006), while the anti-N titer was not significantly elevated in the deceased cases (Appendix A). In fatal cases, the anti-S/RBD titer was higher in the vaccinated patients (*p* = 0.0152), unlike the anti-N titer, which was higher in the unvaccinated patients (*p* = 0.0243).

### 3.4. Cross-Neutralizing Activity against WT and Omicron Variant (BA.1 and BA.4/5) in Patient Groups Based on Vaccination, Infected Variant, and Disease Severity

We assessed neutralizing activities against SARS-CoV-2 WT and Omicron variants (SARS-CoV-2 BA.1 and BA.4/5) using samples collected more than 14 days after infection. When we divided the total patients into vaccinated and unvaccinated infection groups, most patients infected with Delta or Omicron after vaccination displayed a positive neutralizing response to WT (90.0–100.0%), BA.1 (86.7–90.0%), and BA.4/5 (90.0–93.3%) (Figure 4). Interestingly, the neutralization activity against BA.1 and BA.4/5 in the Delta (V) group was comparable to those in the Omicron (V) group.

In the unvaccinated infection group, neutralizing antibody responses against WT were significantly higher in WT and Delta infections compared to Omicron infection (median % inhibition; 88.0%, 92.0%, and 39.0%, respectively). However, neutralizing responses against BA.1 tended to be higher in unvaccinated Omicron infection compared to unvaccinated WT or Delta infection (median % inhibition; 42.0%, 0.8%, and 12.0%, respectively). Unvaccinated Omicron or Delta infections exhibited higher neutralizing activity against BA.1 compared to unvaccinated WT infection (42.0%, 12.0%, and 0.8%, respectively) (*p* < 0.05). Regarding neutralizing activity against BA.4/5, unvaccinated WT, Delta, and Omicron infections showed similar rate of positive responses (55.0%, 61.9%, and 57.1%, respectively) and median % inhibition results (32.8%, 41.0%, and 48.0%) (*p* > 0.05).

Next, we compared the cross-neutralization response in each subgroup based on vaccination and infection strains (Figure 5). Unvaccinated patients with WT and Delta infections exhibited higher neutralizing activity against WT, BA.4/5, and BA.1 strains, in that order (Figure 5A). Vaccinated Delta-infected patients also displayed higher neutralizing activity to WT compared to the Omicron variants (BA.1, *p* = 0.0022, and BA.4/5, *p* = 0.0021), but there was no difference in neutralizing activities between BA.1 and BA.4/5 (*p* > 0.05) (Figure 5B). Concerning Omicron infection, while vaccinated patients exhibited a stronger response, both vaccinated and unvaccinated patients displayed a broader reactivity against WT, BA.1, and BA.4/5 strains. Additionally, we examined the influence of disease severity or the immunocompromised status of individual patients. In unvaccinated patients with WT and Delta infections and vaccinated patients with Omicron infections, there was a trend toward lower cross-neutralizing activity in cases of mild disease and in immunocompromised patients with transplantation and malignancy. However, we were unable to conduct a detailed statistical analysis due to the limited number of patients in each group.

In comparison to non-fatal individuals, neutralizing activity against the WT, BA.1, and BA.4/5 variants all showed a significant elevation in the deceased patient group (WT, *p* = 0.0077; BA.1, *p* = 0.0285; and BA.4/5, *p* = 0.0127) (Appendix A). For the fatal cases, those vaccinated revealed significantly higher neutralizing activity against BA.1 (*p* = 0.0247) compared to the unvaccinated, but not for WT and BA.4/5 (*p* = 0.1456 and *p* = 0.1021, respectively).

## 4. Discussion

The authors investigated the humoral immune response in hospitalized patients, evaluating the kinetics of IgG binding antibodies in the early stages of SARS-CoV-2 infection and assessing whether it is influenced by the disease severity and vaccination. We observed that vaccinated patients displayed significantly higher levels of anti-S/RBD antibodies compared to their unvaccinated counterparts, as reported in previous studies [5,13,19]. Interestingly, patients infected with the Delta variant exhibited significantly higher levels of anti-S/RBD in both unvaccinated and vaccinated cases during early infection, particularly between post-infection days 7 and 21. By post-infection day 21, anti-S/RBD titers were comparable to those induced by other variants. The initial rapid production of anti-S/RBD in patients infected with Delta in both groups suggests that mechanisms beyond the vaccine-induced memory response play a role in the rapid humoral response. This early response may be influenced by the greater severity of Delta variant infections compared to Omicron infections, as previously reported [20,21]. Compared to the differential response caused by the Delta infection, vaccinated individuals demonstrated significantly higher antibody levels even before post-infection day 7, surpassing the rate at which Delta infection could induce. Furthermore, the antibody titer post-day 21 was notably higher in vaccinated individuals. This underscores that vaccines induce a much faster and more substantial production of antibodies compared to the influence a specific variant can exert.

Unlike anti-S/RBD antibodies, no significant difference in anti-N titers was observed between vaccinated and unvaccinated groups, as nucleocapsid was not the target of the vaccines [22]. Furthermore, anti-N antibody levels were not affected by the causative variants in the present study. Similar anti-N levels between vaccinated and unvaccinated patients in both Delta and Omicron infection subgroups support previous reports indicating that vaccination-induced immunity does not elicit an antibody response against the neucleocapsid protein [22,23]. This can be explained by the previous findings that the critical mutations of VOCs are primarily associated with the spike protein [24,25,26]. For these reasons, using anti-N as a correlate of protection is unlikely. Instead, it has been proposed as a marker to guide vaccination decisions [27] or predict severity [28]. Our findings support the use of anti-N in estimating disease severity, as the anti-N titer tended to be higher in severely symptomatic patients, both in unvaccinated and vaccinated individuals.

Previous studies demonstrated the impact of symptom severity on the immune response [11,29]. In this research, we aimed to investigate whether humoral immune response is also influenced by disease severity within predefined subgroups. In line with the studies mentioned earlier, both in the WT and Delta variant infections, patients with severe infections showed higher levels of anti-S/RBD antibodies. The correlation between disease severity and antibody production has also been observed in HIV-1 and Mycobacterium tuberculosis infections [30,31]. This phenomenon is postulated to be driven by a high pathogenic load, prompting B cells to produce more effective antibodies [32].

However, we did not observe this trend among patients infected with the Omicron variant, where higher levels of anti-S/RBD antibodies were also detected in patients with mild disease. To the best of our knowledge, this observation has not been previously reported, and the underlying mechanisms are not yet fully understood. It may be because the immune response to the Omicron variant differs from that of other variants, or it could be influenced by the immunity or comorbidities of hospitalized Omicron-infected patients, leading to distinct patterns of antibody production. Additional research is required to investigate the specific factors contributing to this enhanced antibody response in mildly symptomatic Omicron infections.

When comparing Delta (UV) and Delta (V), as well as Omicron (UV) and Omicron (V), vaccinated groups exhibited higher anti-S/RBD levels compared to their unvaccinated counterparts. The rapid increase in anti-S/RBD levels can be attributed to a robust anamnestic immune response induced by vaccination [33,34].

Finally, we assessed the neutralizing activity against WT, BA.1, and BA.4/5 in each study group. The sVNT was performed using samples collected at least 14 days post-infection, considering the time required for the development of neutralizing activity [35,36,37]. Neutralization to the WT strain was effectively achieved in all five study groups demonstrating 90–100% positivity, except unvaccinated Omicron-infected patients (57.1% positivity). The lower rate of positive response in unvaccinated Omicron infection may be attributed to the occurrence of neutralization escape, as explained previously [9,10]. Conversely, our study showed that vaccinated Omicron-infected patients exhibit increased cross-neutralizing activity against the WT, confirming the effectiveness of vaccination in inducing broader neutralization [38]. In fact, the broader neutralization in vaccinated individuals was demonstrated not only in comparisons between Omicron (UV) and Omicron (V) but also in comparisons between Delta (UV) and Delta (V). Interestingly, vaccinated individuals demonstrated significantly enhanced neutralizing activity against BA.4/5, in contrast to unvaccinated patients. The BA.4/5 subvariants are antigenically more advanced compared to the causative variants of the present study, thus neutralization tests against BA.4/5 help to predict neutralizing activity against newer, emerging variants. This phenomenon can be partly attributed to hybrid immunity, where both vaccination and infection together build a more potent adaptive immune response, in line with previous studies [5,7,33,39]. Furthermore, the vaccine may have broadened neutralization by increasing the number of exposures. As discussed by Luczkowiak, J. et al., repeated antigenic stimuli through infection or vaccination with the wild-type spike are postulated to lead to high avidity and broader neutralization [40]. Specifically, all vaccinated patients in the present study received at least two doses of WT-based vaccines. Following breakthrough infections, each patient experienced more than three exposures to SARS-CoV-2. As the third exposure is postulated to be the most potent in improving the humoral immune response, the number of exposures may play a significant role in the potent and broader neutralization observed in vaccinated individuals.

Notably, the neutralizing activity against BA.1 and BA.4/5 in the Delta (V) group was comparable to that observed in the Omicron (V) group. For a better interpretation of this cross-neutralization, the presence of vaccine-induced memory B (B_mem_) cells should be taken into account, as they contribute to a broader neutralizing activity across different SARS-Cov-2 variants [41]. B_mem_ typically exhibits B cell receptors (BCRs) with relatively low affinity but broad reactivity to heterologous antigens. The B cell repertoire is influenced by antigen exposure and maturation over time [41]. In the case of SARS-CoV-2, antigen exposure drives B_mem_ differentiation and further maturation in germinal centers, leading to broad reactivity to other variants over time. However, this observation cannot be generalized across different viruses, as the direction of B_mem_ response may vary. For instance, repeated homologous flu hemagglutinin vaccination tends to induce a more specific humoral response [42]. Vaccination not only induces B cells but also virus-specific CD4+ T cells, which persist for about four months [43]. These T cells support the generation of high-affinity antibodies and exhibit cross-reactivity to both Delta and Omicron variants [43,44].

When we examined the impact of disease severity and individual immunity on cross-neutralizing activity, as expected, the immunocompromised patients with a history of transplantation and malignancy revealed a trend toward lower cross-neutralizing activity, as did mildly symptomatic patients. Conversely, fatal hospitalized patients exhibited higher production of anti-S/RBD antibodies, as well as more potent and broader neutralization.

There are several limitations to consider in this study. First, the detailed statistical analysis was limited due to the small number of patients in each subgroup. Second, all participants were hospitalized patients in a single center, which could potentially introduce bias into the results. To obtain more robust and generalizable results, further studies with larger sample sizes are necessary. The present study did not examine the influence of racial and ethnic differences in COVID-19 vaccination, which necessitates future comprehensive studies in this area. Additionally, while it has been reported that there are racial and ethnic disparities in the receipt of monoclonal antibody (mAb) treatment among COVID-19 patients, we are unable to address this topic due to none of our patients receiving this treatment during hospitalization. Also, the assessment of neutralizing activity was limited to selected SARS-CoV-2 variants, and not all variants were included, which may compromise the interpretation of cross-reactivity. Additionally, our primary focus was on the short-term humoral response after infection, particularly the measurement of antibody levels. Our previous study in healthy healthcare providers [19] revealed detectable levels of both binding and neutralizing antibodies six months after the second and third dose vaccinations. It remains to be determined whether the observed kinetics hold true for these hospitalized patients and should be further investigated. It is also important to recognize that the long-term response, including T-cell mediated immunity and culture-based neutralizing activity, should be explored in future research.

## 5. Conclusions

It is crucial to acknowledge the ongoing nature of the COVID-19 pandemic and the continuous evolution of the virus. Despite the emergence of new variants, our finding suggest that vaccination based on the ancestral strain still holds value in providing protection against both existing and emerging variants. Therefore, the significance of vaccination in combating the virus remains relevant in the current circumstances. In conclusion, our findings may contribute to the understanding of the humoral immune response against COVID-19, encompassing the impact of variants, disease severity, and vaccination status. Our study suggests the added value of SARS-CoV-2 vaccination in generating more antibodies with broader coverage. Furthermore, our results underscore the benefits of prior vaccination in inducing cross-neutralizing activity against SARS-CoV-2 variants in hospitalized COVID-19 patients, providing valuable insights for vaccination decision-making.

## Figures and Tables

**Figure 1 vaccines-11-01803-f001:**
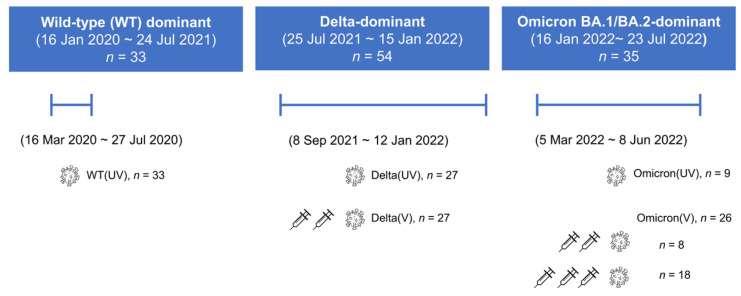
Study enrollment and Determination of SARS-CoV-2 variants. *n*, total number of patients in each group; UV, unvaccinated; V, vaccinated.

**Figure 2 vaccines-11-01803-f002:**
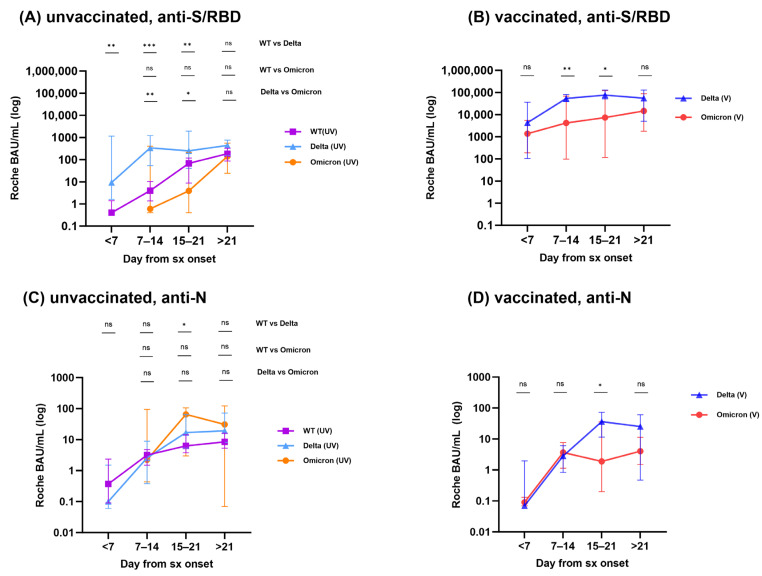
Kinetics of anti-S/RBD and anti-N responses. Kinetics of binding antibody response to WT, Delta and Omicron variants from unvaccinated (**A**,**C**) and vaccinated (**B**,**D**) patients. The median antibody titers of WT, Delta, and Omicron infected individuals are indicated by the squares, triangles, and circles, respectively, and the error bars indicate 95% CI. ns, not significant, * *p* < 0.05, ** *p* < 0.01, *** *p* < 0.001 by Mann-Whitney U test.

**Figure 3 vaccines-11-01803-f003:**
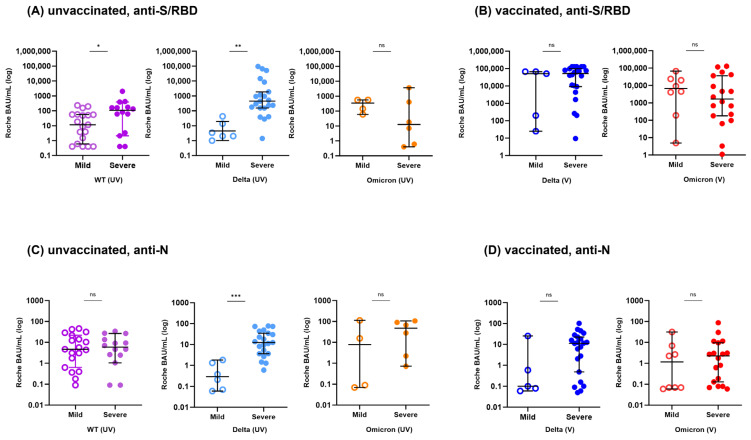
Anti-S/RBD and anti-N antibody levels stratified by disease severity in hospitalized patients infected with SARS-CoV-2 WT, Delta or Omicron variants. Comparison of anti-S/RBD (**A**,**B**) and anti-N (**C**,**D**) antibody levels in patients with vaccinated breakthrough and unvaccinated infections between mild and severe patients with exposure to WT, Delta and Omicron variants, separately. The WT infected group includes only unvaccinated patients. Each group is color-coded as follows: WT (UV), purple; Delta (UV), light blue; Omicron (UV), orange; Delta (V), blue; Omicron (V), red. The median antibody titers are indicated by the horizontal lines and the error bars indicate 95% CI. ns, not significant, * *p* < 0.05, ** *p* < 0.01, *** *p* < 0.001 by Mann-Whitney U test.

**Figure 4 vaccines-11-01803-f004:**
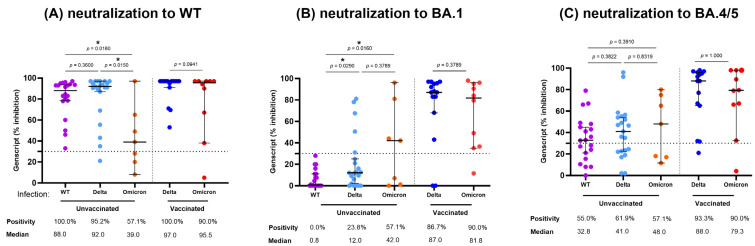
Neutralizing activity to Wild type and Omicron variants (BA.1 and BA/4/5) in hospitalized patients infected with SARS-CoV-2 WT, Delta or Omicron variants. Neutralizing activity against Wild type (WT) (**A**), Omicron BA.1 (**B**) and Omicron BA.4/5 (**C**) in each vaccine and variant combination. Each group is color-coded as follows: WT (UV), purple; Delta (UV), light blue; Omicron (UV), orange; Delta (V), blue; Omicron (V), red. The horizontal dotted line represents the cutoff of the assay (30%). Median percent inhibition of neutralizing antibodies is indicated by the horizontal lines and the error bars indicate 95% CI. ns, not significant, * *p* < 0.05, by Mann-Whitney test.

**Figure 5 vaccines-11-01803-f005:**
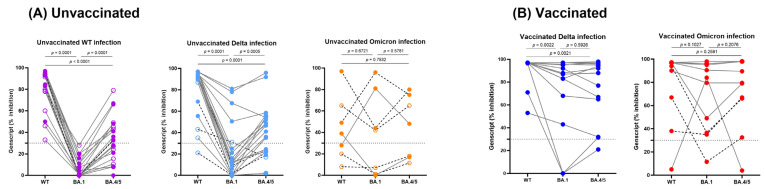
Comparison of neutralizing activity to Wild type and Omicron variants (BA.1 and BA.4/5) in unvaccinated (**A**) and vaccinated (**B**) patients with SARS-CoV-2 WT, Delta or Omicron variants. The horizontal dotted lines represent the cutoff of the assay (30%). Each group is color-coded as follows: WT (UV), purple; Delta (UV), light blue; Omicron (UV), orange; Delta (V), blue; Omicron (V), red. Dotted line connecting points indicated immunocompromised patients. Empty circle indicates patients with mild disease and full circle indicates patients with severe disease.

**Table 1 vaccines-11-01803-t001:** Characteristics of the included patients.

	WT Infection (*n* = 33)	Delta Infection (*n* = 54)		Omicron Infection (*n* = 35)	
	Unvaccinated (*n* = 33)	Unvaccinated (*n* = 27)	Vaccinated (*n* = 27)	Unvaccinated (*n* = 9)	Vaccinated (*n* = 26)
Age *	60 (50–70)	69 (57–81)	74 (65–82)	67 (55–82)	70 (62–77)
Female, *n* (%)	17 (51.5%)	10 (37.0%)	10 (37.0%)	4 (44.4%)	9 (34.6%)
No medical risk, *n* (%)	9 (27.3%)	3 (11.1%)	3 (11.1%)	1 (11.1%)	1 (3.8%)
Any risk present, *n* (%)	24 (72.7%)	24 (88.9%)	24 (88.9%)	8 (88.9%)	25 (96.2%)
Obesity (BMI > 30 kg/m^2^)	0	0	1	0	0
Smoking ^†^	1	1	1	0	2
Hypertension	11	21	14	3	8
Diabetes mellitus ^‡^	3	7	11	4	8
Malignancy	3	3	1	4	8
Cerebrovascular disease	2	2	1	0	2
Chronic kidney disease	2	1	2	0	2
Chronic lung disease	1	0	2	0	1
Tuberculosis	1	2	1	1	2
Heart conditions ^§^	0	1	3	2	5
Dementia	0	4	4	0	0
Transplantation recipient ^∥^	0	2	0	1	0
COVID-19 vaccination history					
3 doses	NA	NA	0	NA	18 **
2 doses	NA	NA	27 ^¶^	NA	8 ^††^
Unvaccinated	33	27	NA	9	NA
COVID-19 disease severity					
Mild	19 (57.6%)	6 (22.2%)	5 (18.5%)	3 (33.3%)	8 (30.8%)
Severe	14 (42.4%)	21 (77.8%)	22 (81.5%)	6 (66.7%)	18 (69.2%)
In-hospital fatalities	1	7	7	3	11
Number of samples, *n*	120	90	84	20	55
Days from symptom onset to sample collection	2–40	1–45	1–41	7–54	1–39
<7	13 (10.8%)	16 (17.8%)	17 (20.2%)	0 (0.0%)	22 (40.0%)
7–14	48 (40.0%)	36 (40.0%)	44 (52.4%)	9 (45.0%)	15 (27.3%)
15–21	28 (23.3%)	20 (22.2%)	14 (16.7%)	4 (20.0%)	5 (9.1%)
>21	31 (25.8%)	18 (20.0%)	9 (10.7%)	7 (35.0%)	13 (23.6%)

Abbreviation: WT, wild type; BMI, body mass index; COVID-19, coronavirus disease 2019; NA, not applicable. * median age in years (interquartile range); ^†^ both current and former smokers included; ^‡^ both type 1 and type 2 diabetes mellitus patients included; ^§^ such as heart failure, coronary artery disease, or cardiomyopathies; ^∥^ solid organ or hematopoietic cell transplantation; ^¶^ 12 patients had completed ChAdOx1 (AstraZeneca) primary series, seven patients had completed BNT162b2 (Pfizer BioNTech) primary series, and one patient had received a heterologous combination of BNT162b2 and mRNA-1273 (Moderna). The remaining seven patients reported having received two doses of vaccine, but the vaccine platform was not identified in detail.; ** The vaccine type was not clarified in each patient, and five of the patients were reported to have received booster vaccination based on the mRNA platform (i.e., BNT162b2 or mRNA-1273).; ^††^ Five of the patients reported at least one dose of BNT162b2, otherwise unspecified.

## Data Availability

The data presented in this paper are available on request from the corresponding author.

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
