# Peer review of "Humoral Response Kinetics and Cross-Immunity in Hospitalized Patients with SARS-CoV-2 WT, Delta, or Omicron Infections: A Comparison between Vaccinated and Unvaccinated Cohorts"

_vaccines, 2023, doi:10.3390/vaccines11121803_

Round 1

Reviewer 1 Report

Comments and Suggestions for Authors

The paper analyzes the humoral immune response to spike and NP in hospitalized COVID-19 patients, evaluating the role of previous vaccinations, SARS CoV 2 variants and disease severity.

For any virus variants, vaccinated subjects had higher anti RBD (but not anti NP) antibody titers. Delta variant elicited the highest anti RBD and anti NP antibody titers. As previously reported, severe cases had the highest antibody titer, with the exception of patients infected by Omicron variant.

Of interest, crossneutralization was observed following vaccination and infection with any of the variants.

The paper reports new data, adequately presented and discussed.

Only a minor observation can be raised: no data on the outcome of the infection are given.

There were fatal  infections? In case, what parameters of the immune reponse were related to fatal outcome?

Reviewer 2 Report

Comments and Suggestions for Authors

The emergence of SARS-CoV-2 has led to a global pandemic with a significant impact on public health. Vaccination has been a crucial strategy in combating the spread of the virus and reducing the severity of COVID-19, the author conducted a comprehensive analysis to examine the effects of previous vaccination, the causative variant and disease severity on the kinetics of humoral immune response in a specific subgroup of hospitalized COVID-19 patients. Additionally, the author evaluated the ability to generate cross-neutralizing activity against both the original SARS-CoV-2 strain and the Omicron variant. However, there are still some concerns not well clearly addressed, the comments are as follow:

Major concern:

1.      Previous published references, such as Molecular Diagnosis & Therapy (2023) 27:159–177, Viruses 2023 Sep 26;15(10):1994. Suggest that variation in the SARS-CoV-2 virus exhibits significant variability in severity. Thus, we would like to see neutralization to the newest variants.

2.      How long neutralization activity persists in individuals in South Korea.

3.      The author failed to discuss the racial and ethnic differences in COVID-19 vaccination.

4.      Racial and ethnic disparities in receipt of monoclonal antibody (mAb) treatment among patients with COVID-19, therefore, the author should provide a comparison with or without mAb treatment and vaccinated patient.

5.      In figure 2, all y axes should be revised as log.

6.      Why did not investigate the neutralizing activities against SARS-CoV-2 WT and Omicron variants BA.2 and BA.3.

Reviewer 3 Report

Comments and Suggestions for Authors

The work by Kang et al. focused on features the effects of previous vaccination, the causative variant and disease severity on the kinetics of humoral immune response in patients with COVID-19 by looking at the levels of antibodies with different specificity, as well as the virus neutralization tests. Though it exists already several data in the literature, divergent results have been published and this new set of data can expand our knowledge on the current pandemic. Though the data presented here are very interesting, I have a few comments that I hope would help improved the manuscript.

Line 77. "...  as described in a previous study [12]." The description should be add. Something like " In brief, ..."

Line 108. Similarly, the description should be add. Something like " In brief, ..."

I think that the literary data on SARS-CoV-2-specific memory B cells and follicular Th cells also should be added to 'Discussion'.

I find that the authors could have discusses more in the 'Discussion' section. It is lacking of interpretations and/or hypothesis.  Are the obtained results consistent with what has been seen for other viral infections (for instance, influence virus infection).

Comments on the Quality of English Language

Minor editing of English language required.

Reviewer 4 Report

Comments and Suggestions for Authors

The results presented by the authors are relevant and quite clearly illustrated. The work is of certain scientific interest and may be useful to clinical specialists. I have no significant comments to the authors.

Author Response

The authors extend our gratitude for your keen and thoughtful comment.